# A Refined Approach to Permanent Coronary Artery Ligation in Rats: Enhancing Outcomes and Reducing Animal Burden

**DOI:** 10.3390/ani16010099

**Published:** 2025-12-29

**Authors:** Ellen Heeren, Lotte Vastmans, Dorien Deluyker, Marc Hendrikx, Virginie Bito

**Affiliations:** Cardio & Organ Systems (COST), Biomedical Research Institute, UHasselt—Hasselt University, Agoralaan, 3590 Diepenbeek, Belgium; ellen.heeren@uhasselt.be (E.H.); lotte.vastmans@uhasselt.be (L.V.); dorien.deluyker@uhasselt.be (D.D.); marc.hendrikx@uhasselt.be (M.H.)

**Keywords:** permanent ligation, myocardial infarction, rat model, 3Rs, surgical refinement, reproducibility

## Abstract

Rodent models continue to be used for studying heart attacks and developing new therapies. However, the standard surgical method to induce a heart attack in rats, achieved by permanently tying off a coronary artery, often causes high mortality and variable results. We refined this procedure to increase survival outcomes, reproducibility, and to reduce animal burden. The refinements include improved surgical access to the heart while avoiding lung injury, continuous monitoring of temperature and heart rate, and timely adjustments of retractors and anesthetic depth to prevent cardiac complications. With these improvements, we eliminated surgical mortality and achieved consistent heart attacks, thereby increasing both scientific reliability and animal welfare.

## 1. Introduction

To investigate disease mechanisms and evaluate novel therapeutic strategies for myocardial infarction (MI), a growing number of non-animal approaches have emerged in recent years. These include ex vivo studies using human myocardial tissue, as well as advanced in vitro platforms designed to model specific aspects of cardiac pathology [1]. While access to human myocardial samples is inherently limited, recent technological advances have enabled increasingly sophisticated use of these scarce materials [2]. Among in vitro approaches, cardiomyocytes derived from human induced pluripotent stem cells (hiPSC-CMs) have proven valuable for investigating cellular pathways involved in cardiovascular disease and for screening potential therapeutics [3]. To better approximate the complexity of native cardiac tissue, multicellular systems such as three-dimensional cardiac organoids have been developed, incorporating cardiomyocytes, fibroblasts, endothelial cells, and other supporting cell types, such as adipose-derived stem cells [4]. Despite these advances, current in vitro and ex vivo models remain limited in their ability to capture the full complexity of myocardial infarction. In particular, they do not adequately reproduce the dynamic multicellular interactions, neurohumoral regulation, mechanical loading, or systemic responses such as immune activation and remodeling processes that critically shape post-infarction outcome [5]. As a result, in vivo rodent models continue to be frequently employed to study myocardial infarction and its progression toward heart failure (HF). Rodent models are widely used in preclinical MI research with both mice and rats offering distinct advantages. Mice are extensively employed due to their well-characterized genome and the broad availability of genetically modified strains [6]. However, their small heart size and high heart rates (300–800 beats per minute) com-plicate intricate surgical procedures and cardiac imaging. These challenges are less pronounced in rats, which possess a myocardial mass 10-fold larger than that of mice and a lower heart rate (250–500 beats per minute), facilitating surgical manipulation and functional assessment [6,7]. While anatomical and electrophysiological differences compared to humans must be acknowledged, rat models allow investigation of myocardial infarction within an intact, living system, capturing complex tissue-level and systemic responses that are not fully replicated in vitro or ex vivo [8]. The permanent ligation (PL) of the left anterior descending coronary artery (LAD) is an extensively used method to induce MI in rodents described in the literature, with numerous protocols available [9]. Nevertheless, this method remains technically demanding and invasive, with reported perioperative mortality rates reaching up to 50% in rats and considerable variability in infarct size and severity [10,11]. Mortality typically arises from arrhythmias induced by ischemia, bleeding, and pneumothorax or respiratory complications related to surgical invasiveness or accidental tissue injury [12]. These factors negatively affect reproducibility, data quality, and animal welfare.

Although substantial progress has been made toward non-animal alternatives, complete replacement of in vivo myocardial infarction models remains challenging due to the complexity of the disease process [5]. Within this context, reduction and refinement represent particularly important and immediately actionable strategies within the 3Rs framework first articulated by Russell and Burch [13]. High perioperative mortality and procedural variability inherent to the rat PL model often necessitate the use of large animal numbers to achieve sufficient statistical power. While anatomical variability of the coronary vasculature limits the extent to which infarct variability can be eliminated, perioperative mortality represents a modifiable outcome [14,15]. Reducing surgical mortality therefore directly contributes to animal reduction by decreasing the number of animals required to obtain valid experimental datasets. Despite the abundance of published PL protocols, many provide limited insight into the critical procedural steps that most strongly influence surgical outcome and perioperative survival and experimental success [16]. Refinement of the PL procedure can be achieved through optimization of perioperative care and surgical technique, with a particular focus on minimizing invasiveness and procedure-related complications. Standardized airway management, for example, is especially critical, as respiratory complications are a major cause of mortality in early PL protocols. While orotracheal intubation, rather than tracheotomy, is increasingly used in rodent surgery, technical challenges and airway trauma remain significant risks [11,17,18]. Together with strict temperature control and careful tissue handling, such refinements aim to improve perioperative stability and survival without altering the fundamental characteristics of the model.

In this study, we present a refined rat LAD PL protocol that integrates standardized airway management, strict temperature control, and surgical technique refinements, including the use of a reference ligature for reliable identification of the LAD. By systematically evaluating perioperative outcomes across successive experimental cohorts, we identify key procedural factors contributing to perioperative mortality and demonstrate how targeted refinements can effectively mitigate these risks. This approach resulted in a marked reduction in mortality and improved reproducibility of infarct induction, supported by functional and histological validation. Collectively, this work provides practical guidance for refining an established model in a manner that enhances animal welfare and experimental reliability.

## 2. Materials and Methods

### 2.1. Equipment Setup and Preparation

Prior to surgery, all instruments were tip-sterilized using a hot bead sterilizer (Thermo Fisher Scientific, Waltham, MA, USA) and arranged on a sterile field according to the order of use. The VentStar Small Animal Ventilator (RWD Life Science, San Diego, CA, USA) was set to volume mode with a respiratory rate of 55 breaths per minute, tidal volume of 2 mL, peak inspiratory pressure of 35 cmH_2_O, and positive end-expiratory pressure of 2 cmH_2_O. A heating pad was placed beneath a silicone mat (René Remie Surgical Skill Centre, Almere, The Netherlands) to maintain the animal’s body temperature during the procedure. 25 G needles (Terumo, Leuven, Belgium) served as anchor points at the corners of the mat to facilitate the attachment of silicone-coated retracting hooks (René Remie Surgical Skill Centre, Almere, The Netherlands). The intubation stand was positioned under a stereomicroscope (Wild Heerbrugg, Heerbrugg, Switzerland) and connected to an isoflurane delivery system (Eickemeyer, Tuttlingen, Germany). Snake lights (Avantor, Radnor, PA, USA) were arranged to illuminate the rat’s throat, improving visualization during intubation.

### 2.2. Anesthesia and Intubation

Female Sprague Dawley rats (Janvier Labs, Le Genest-Saint-Isle, France) of 8 weeks old were used for all experiments. A total of 172 rats were used across five cohorts. Solely female rats were used to ensure consistency across cohorts and to reduce sample size. Animals were weighed to calculate the buprenorphine dose (0.04 mg/kg; Bupaq, Richter Pharma AG, Wels, Austria) prior to induction. Anesthesia was induced in an induction chamber with 5% isoflurane in oxygen (2 L/min) until a surgical plane characterized by suppression of the gag reflex was reached. Respiration was continuously monitored visually and via the ventilator parameters, while heart rhythm was assessed visually throughout the procedure to prevent anesthesia-related complications. Following induction, rats were positioned on the intubation stand (René Remie Surgical Skill Centre, Almere, The Netherlands), securing the upper incisors to stabilize the head. Orotracheal intubation was performed under microscopic visualization using direct throat illumination, gentle tongue retraction with a laryngoscopic blade (René Remie Surgical Skill Centre, Almere, The Netherlands), and slight dorsal displacement of the trachea by an assistant. When visualization was limited, the endotracheal tube was carefully advanced to open the oropharynx and expose the vocal cords, allowing clear identification of the trachea prior to tube placement. A 16 G endotracheal tube (Venflon, Becton Dickinson, Franklin Lakes, NJ, USA) was inserted into the trachea, with correct placement verified by condensation (“fogging”) on a microscope slide (Epredia, Kalamazoo, MI, USA) held above the tube opening during ventilation. Following successful intubation, animals were transferred to a heating pad and connected to the ventilator. Once spontaneous breathing over the ventilator ceased, the isoflurane concentration was reduced to 2.5% and oxygen flow to 1 L/min to maintain an appropriate anesthetic depth. Forelimbs were loosely secured, and a rectal probe (FUJIFILM VisualSonics Inc., Toronto, ON, Canada) was inserted to monitor core body temperature, which was maintained between 35 and 38 °C throughout the procedure. The heating pad was adjusted as needed to prevent hypothermia or hyperthermia, particularly during prolonged surgeries or post-ligation recovery. This standardized setup allowed reliable airway management under exclusive inhalation anesthesia while minimizing the risk of esophageal intubation or airway trauma (Figure 1A–I).

### 2.3. LAD Ligation

The rat’s chest was shaved and depilated. The surgical field was disinfected three times by alternating a povidone–iodine (Betadine^®^, Mundipharma, Cambridge, UK) and 70% ethanol. Ophthalmic gel (Vidisic^®^ carbomerum 980 2 mg/g, Bausch + Lomb, Laval, QC, Canada) was applied to prevent corneal drying, and analgesia was administered intramuscularly in the hind leg with the precalculated dose of buprenorphine (0.04 mg/kg). A sterile adhesive drape (Glad^®^ Press ’n Seal, Glad, Oakland, CA, USA) was positioned over the thorax to maintain asepsis, whereafter a small surgical window was created to expose the surgical field. A longitudinal midline skin incision (~1 cm) was made over the sternum, approximately 5 mm caudal to the base of the left axilla. Subcutaneous tissue was bluntly dissected to create a lateral pocket extending toward the axilla, exposing the *Pectoralis major* muscle. The *Pectoralis major* was gently separated from the underlying intercostal muscles and displaced laterally to the right side of the animal, where it was secured using retracting hooks to provide stable exposure. Excessive traction was avoided to prevent displacement of the left lung over the ventricle, which could obstruct visualization and increase the risk of pulmonary injury. The third intercostal space was identified by alignment with the base of the left axilla, and a left-sided thoracotomy was performed by blunt dissection of the intercostal muscles near the sternum. The opening was gently expanded to approximately 4 mm^2^. A medial entry point was selected to reduce the likelihood of pulmonary protrusion into the surgical field while allowing visualization of the internal thoracic (mammary) artery, which was carefully avoided or gently displaced using a blunt retracting hook. Correct entry at the appropriate intercostal level was verified by partial visualization of thymic tissue and the underlying ventricle. The pericardium was opened using micro scissors, taking care to avoid damage to the thymus or surrounding tissues. When necessary, the thymus was gently elevated and displaced cranially using a foam swab to improve visualization. The left atrial appendage and pulmonary conus served as anatomical landmarks for identifying LAD. The LAD was ligated approximately 2–3 mm below the left atrial appendage using an 8-0 Prolene suture (Ethicon Inc., Raritan, NJ, USA). Successful ligation was confirmed by immediate pallor of the left ventricular wall distal to the ligation site, indicating effective coronary occlusion. In cases of incomplete or patchy discoloration, suggesting partial or unsuccessful ligation, an additional ligature was placed medially or laterally. The initial ligature was retained as a reference point, with its standing ends left slightly longer to facilitate gentle manipulation of the heart and assist in positioning subsequent ligatures. Mild upward traction on the reference ligature was occasionally used to improve visualization of the LAD. Following confirmation of coronary occlusion, the isoflurane concentration was reduced to 1% to stabilize cardiac rhythm. The thoracic cavity was inspected for bleeding, and any residual blood was removed using a foam swab. The intercostal muscles were closed using 5-0 Vicryl sutures (Ethicon Inc., Raritan, NJ, USA) while gently compressing the rib cage to expel intrathoracic air and prevent pneumothorax. The *Pectoralis major* was repositioned, and the skin was closed with interrupted sutures. The wound was disinfected with povidone–iodine prior to recovery. Throughout the procedure, body temperature was maintained between 35 and 38 °C, and anesthetic depth was adjusted as needed to ensure stable respiration and heart rhythm. During prolonged procedures, particular attention was paid to the positioning and tension of retracting hooks to avoid restriction of cardiac movement, which could otherwise induce bradycardia. The surgical procedure is illustrated in Figure 2.

### 2.4. Post-Operative Care

Following completion of the surgical procedure, isoflurane was discontinued while maintaining oxygen flow at 1 L/min. The surgical drape and the tape securing the forelimbs and endotracheal tube were carefully removed. A surgical jacket was placed on the animal while it remained anesthetized (Appendix A). Recovery was closely monitored by assessing reflex responses (toe pinch) and spontaneous movements of the whiskers and tail. Extubation was performed only once spontaneous breathing resumed and voluntary movements were observed to prevent respiratory arrest. When uncertain, ventilation was briefly paused to confirm independent respiration. Extubation occurred before the onset of intense limb movements to avoid tracheal injury. As the animal regained consciousness, the surgical jacket was secured with tape to prevent the animal from disturbing the sutures or incision. Animals were then transferred to a clean type IV cage containing food, water, and enrichment materials. To prevent wound interference, rats were housed individually for 24 h, and cages were clearly labeled with surgical and identification details. If the jacket remained in place after 24 h, it was carefully removed and sutures were inspected. Animals with intact sutures were reunited with their cage mates, whereas those requiring re-suturing were housed individually for an additional 24 h. Buprenorphine (0.04 mg/kg, intramuscular) was administered 24 h post-surgery for analgesia. Postoperative monitoring included daily body weight measurements and visual inspection of the incision site for seven days to assess recovery and wellbeing.

### 2.5. Data Collection and Statistics

Data were collected from five cohorts (cohort 1: *n* = 37, cohort 2: *n* = 25, cohort 3: *n* = 73, cohort 4: *n* = 20, cohort 5: *n* = 17), with each surgical procedure documented in real time using a standardized spreadsheet. Parameters included survival outcome, intra- and post-operative mortality, and suspected cause of death (if applicable). Intraoperative mortality was defined as the percentage of animals that died during surgery relative to the total cohort size. Post-operative mortality was defined as death within 7 days post-surgery and expressed as the percentage of animals that died after recovery from anesthesia. When permitted by the experimental design, MI was confirmed by echocardiography (Vevo^®^ 3100, FUJIFILM Visualsonics, Inc., Toronto, ON, Canada) seven days after ligation (cohorts 3–5). MI verification was omitted in cohorts 1–2 due to immediate post-ligation therapeutic interventions that could have biased infarct assessment. In cohort 3, only animals from batches performed under standard protocol conditions were included in the success rate calculation, whereas those with technical deviations or intentional procedural variations were excluded. Surgical success rate was calculated as the number of animals with confirmed MI divided by the number of surviving animals. For learning curve analysis, the number of animals required by two independent operators to achieve consistent ligations was recorded and compared descriptively. Cohort 5 was used for MI characterization, as this group was operated using the fully optimized and refined surgical protocol. Echocardiography was conducted to determine left ventricular ejection fraction (LVEF) using parasternal long-axis B-mode images. Infarct extent and severity were quantified using a 16-segment wall motion scoring model, from which the wall motion score index (WMSI) was calculated by summing all segmental scores and dividing by 16. Infarct size (as a percentage) was determined by dividing the number of segments with a score > 2 by 16 and multiplying by 100. For histological confirmation, cardiac tissue was collected and processed for Sirius Red staining, allowing visualization of infarcted myocardium (red) and viable myocardium (green). Statistical analysis were performed in Graphpad Prism (v10.6.1, GraphPad Software, San Diego, CA, USA). A Wilcoxon signed-rank test was performed, with data expressed as median and interquartile range (IQR). A non-parametric approach was chosen given the paired design and limited sample size (*n* = 15). Statistical significance is defined as *p* < 0.05.

## 3. Results

### 3.1. Refined Surgical Approach for Permanent LAD Ligation

#### 3.1.1. Orotracheal Intubation Procedure

Application of the refined orotracheal intubation setup enabled rapid and reproducible airway access under inhalation anesthesia across all experimental cohorts, with no observed cases of esophageal intubation or intubation-related complications. The standardized visualization approach facilitated consistent and swift intubation and stable ventilation throughout surgery (Figure 1).

#### 3.1.2. LAD Ligation Procedure

Surgical refinements to the thoracotomy and LAD ligation procedure consistently improved exposure of the left ventricle and LAD while minimizing interference from surrounding tissues. The use of a medial thoracotomy and reference ligature facilitated reliable LAD identification and correction of initial failed ligation attempts (Figure 2). Collectively, these refinements reduced procedure-related complications and improved reproducibility across operators.

### 3.2. Refined PL Protocol Improves Surgical Outcome

Progressive refinements of the PL procedure greatly improved surgical outcomes across experimental cohorts (Figure 3). In the initial cohort, intraoperative and postoperative mortality rates were 27% and 4%, respectively, when using a standard protocol. With successive protocol optimizations, mortality rates declined steadily, reaching 0% in the final cohort (Figure 3A). Although direct visual identification of the LAD was possible in only 48% of cases, the introduction of a reference ligature resulted in a mean infarction success rate of 94.3% (Figure 3B). Causes of mortality were primarily related to respiratory complications, accidental cardiac puncture, and cardiac arrest in early cohorts, all of which were eliminated following refinements (Figure 3C). Training outcomes demonstrated a pronounced difference between operators, with an operator trained under supervision using the optimized procedure requiring approximately 59% fewer practice animals to achieve consistent surgical success compared to an independently trained operator (Figure 3D). These data demonstrate that cumulative protocol refinements directly eliminated the major causes of perioperative mortality observed in early cohorts.

### 3.3. MI Characterization

Echocardiography was employed to determine LVEF from parasternal long-axis B-mode images, revealing a significant reduction four weeks post-MI compared to baseline (median 40.7% [IQR 34.6–49.2%] vs. 64.1% [IQR 61.7–68.9%] *p* < 0.0001) (Figure 4A). Infarct extent and severity were further assessed using the 16-segment wall motion scoring model (Figure 4B). In line with the LVEF, WMSI showed significantly increased wall motion abnormalities at four weeks post-MI compared to baseline (median 1.50 [IQR 1.30–1.70] vs. 1.00 [IQR 1.00–1.00] *p* < 0.0001). Estimated infarct size corresponded to a median of 31.0% [IQR 25.0–38.0%] of the left ventricle at four weeks post-MI (Figure 4C). Regional wall motion abnormalities were visualized using a 16-segment bull’s-eye plot of the left ventricle. The plot revealed hypo- to akinetic regions predominantly involving the mid and apical anterior and anterolateral segments four weeks post-MI (Figure 4D). Lastly, collagen deposition and infarct formation were visualized using Sirius Red staining in a transverse section at the level of the papillary muscles. Healthy myocardium appeared green, whereas fibrotic scar tissue was stained red, illustrating the extent of post-MI remodeling (Figure 4E).

## 4. Discussion

To date, a clinically translatable therapy for successful myocardial repair following MI has yet to be established [19]. As ischemic heart disease remains the leading cause of death worldwide, with HF representing a major long-term consequence of MI and a leading cause of morbidity and hospitalization, continued reliable research is warranted [20,21]. This underscores the importance of carefully designed and ethically refined experimental models when in vivo approaches are employed. This study describes a rigorously optimized rat PL protocol, designed to reliably induce MI while eliminating perioperative mortality. While numerous variations in the rat LAD PL model have been described, most reports primarily focus on infarct size, functional outcomes, or post-infarction remodeling, with perioperative mortality reported as a secondary outcome or accepted as an unavoidable limitation. In contrast, the present study explicitly treats perioperative mortality as a modifiable experimental variable rather than an inherent feature of the model. By evaluating outcomes across successive experimental cohorts and systematically integrating refinements in anesthesia, airway management, thoracic access, coronary identification, temperature control, and operator training, we demonstrate that perioperative mortality in the rat PL model can be effectively eliminated. This represents a shift from isolated technical adjustments toward a standardized, systems-level optimization of the procedure, with direct implications for reproducibility, animal welfare, and experimental efficiency.

Depending on operator expertise and the specific protocol applied, mortality rates range from as low as 5% in protocols where the heart is exteriorized for ligation, to 50% with more classical approaches [22,23]. Although mortality rates are generally reported transparently in the literature, the underlying causes are rarely discussed in detail [16]. Arrhythmias resulting from large infarcts are often cited as a main cause of mortality [24]. Yet many additional perioperative risks, such as respiratory failure, hypothermia, mechanical restriction of cardiac movement and airway trauma, remain underestimated and poorly addressed in an integrated manner. Several studies have described individual refinements to the rat LAD ligation procedure, including alternative anesthetic regimens, thoracotomy approaches, or modified ligation strategies [25,26]. However, these refinements are typically implemented in isolation and rarely evaluated for their cumulative impact on perioperative mortality. In the present work, no single modification alone was sufficient to achieve the observed improvement in survival. Rather, elimination of perioperative mortality resulted from the coordinated implementation of multiple, interdependent refinements addressing distinct yet converging risk factors. This highlights the multifactorial nature of perioperative mortality in the PL model and explains why isolated procedural modifications reported in earlier studies have not consistently translated into uniformly low mortality across laboratories.

Standardized airway management was a key component of this integrated refinement strategy. The use of a dedicated intubation setup allowed for exclusive isoflurane-based inhalation anesthesia and eliminated the need for supplemental injectable agents, such as xylazine/ketamine, which are associated with increased anesthetic-related complications [12,14]. While volatile anesthetics have been reported to exert cardioprotective effects through preconditioning mechanisms via modulation of mitochondrial function and cell-survival pathways, the present study was not designed to evaluate the underlying mechanisms for cardioprotection [27]. Nevertheless, the standardized use of isoflurane combined with controlled ventilation and oxygen delivery likely contributed to improved perioperative stability. Importantly, the benefit of this approach lies not in anesthetic choice alone, but in its integration with reliable airway control, continuous monitoring, and carefully timed extubation, thereby minimizing respiratory complications that frequently contribute to early mortality.

Surgical access to the heart represents another major determinant of outcome. Left thoracotomy inherently carries a risk of pulmonary injury, which can be minimized during mobilization of the *Pectoralis major* to expose the intercostal musculature. Excessive lateral displacement can allow the left lung lobes to slide over the left ventricle and encroach on the surgical field. Accordingly, positioning the animal in strict dorsal recumbency and creating the intercostal incision medially, adjacent to the sternum between the third and fourth ribs, mitigates this risk, but also exposes the internal thoracic (mammary) artery [14]. While initially seeming to pose a hazard, the artery can be safely displaced with blunt hooks, minimizing the risk of accidental puncturing or nicking. In rare cases where the left lung lobes still interfere, gentle displacement using a piece of foam to protect the lung tissue safely restores unobstructed visualization. Careful retraction of the thymus with a protective foam barrier further minimized tissue trauma and maintained and unobstructed surgical field. Collectively, these refinements effectively eliminated respiratory-failure-related deaths observed in cohort 1.

With the surgical field adequately prepared and the heart exposed, attention then shifts to the primary objective, namely successful ligation of the LAD. In contrast to larger mammalian species, the LAD in rats runs intramurally, which can make it difficult to clearly distinguish [14,16]. In our hands, we found the LAD to be visible in only 48% of cases on average. However, the introduction of a reference ligature to guide lateral placement of subsequent ligations provided a practical solution to this limitation, resulting in a 94% success rate. Throughout the procedure, continuous monitoring of cardiac rhythm enables rapid detection of bradycardia or arrhythmias. When these occur prior to the successful ligation, they often reflect restricted cardiac motion due to clotted blood or retractor tension. Prompt hemostasis and adjustment of retractors typically restored function, with transient reduction in isoflurane providing an additional safeguard. With particular attention to subtle changes in heart rate and rhythm, this approach has effectively minimized procedural mortality, allowing us to achieve zero perioperative mortality in the final optimized cohort. Finally, properly timed extubation is critical, as premature extubation may lead to respiratory arrest, whereas delayed extubation can cause tracheal injury. We have identified the optimal extubation window occurring once the animal is nearly awake, showing movement in the whiskers, limbs, or tail.

Maintenance of normothermia emerged as another critical determinant of peri-operative survival as it is known to have a significant impact on survival rates [11,28]. Rodents are particularly prone to hypothermia during anesthesia, which can adversely affect heart rate, coagulation, infarct size and cardiac function in general [29]. Continuous temperature monitoring combined with active warming using heating pads and external warming devices are routinely employed. In addition, the type of surgical drape used also plays a role in maintaining normothermia. In our protocol, we use Glad^®^ Press ‘n Seal cling film as surgical drape, which is suitable for aseptic procedures and helps prevent body temperature loss during anesthesia [29,30]. Based on our experience, maintaining body temperature within the range of 35–38 °C is optimal for survival, whereas failure to keep animals within this range was associated with higher mortality [31]. Continuous intraoperative monitoring with a rectal probe is therefore essential to significantly reduce mortality rates.

Beyond procedural refinements, the PL procedure is associated with a substantial learning curve that represents an often-overlooked source of animal use. Importantly, this study quantitatively demonstrates that structured, supervised training using an optimized protocol significantly reduces the number of animals required for operators to achieve consistent surgical success. Indeed, extensive training is needed to identify the LAD and perform the ligation successfully without damaging surrounding tissues or causing severe bleeding. Ideally, new operators should be trained under the supervision of an experienced operator who can provide real-time feedback. This is particularly important given the variability in coronary artery anatomy, which requires operators to become familiar with a range of anatomical scenarios [16]. Operators trained under guidance required approximately 59% fewer animals compared to independent training, highlighting that refinement and reduction extend beyond surgical technique alone. Addressing operator-dependent variability through standardized training therefore constitutes an additional and highly effective reduction strategy within the 3Rs framework.

While intraoperative ECG recordings were not collected, infarct induction and functional consequences were validated using longitudinal echocardiographic assessment and histological analysis, which together provide robust confirmation of myocardial injury and remodeling. With this refined protocol, we were able to consistently induce MI. Echocardiographic analysis showed a marked reduction in LVEF (median 40.7%), which falls within the range typically reported after MI in similar models [32]. To better capture infarct extent and severity, we also assessed WMSI as an indicator of regional wall motion abnormalities [33]. In line with the LVEF results, WMSI revealed wall motion abnormalities in the mid and apical anterior and anterolateral segments, corresponding to the expected perfusion area of the LAD [34]. The estimated infarcted myocardial fraction derived from WMSI segments had a median of 31% [IQR: 28–35%], showing that even with procedural refinements, some variability in infarct size remains. Although some variability in infarct size persisted, this likely reflects inherent individual anatomical differences rather than procedural inconsistency. This is in agreement with the findings of Kainuma et al., who reported that, despite consistent LAD ligation below the left atrial appendage by experienced operators, about one-third of rats developed smaller infarcts limited to the anterior wall. Their study further linked this difference to variability in coronary branching patterns, which can influence the territory affected by occlusion [35]. Overall, these findings emphasize that while infarct variability cannot be entirely eliminated, perioperative mortality can be effectively addressed through procedural refinement. Nonetheless, careful consideration of infarct size variability remains important for both experimental design and data interpretation.

Careful attention to the critical surgical steps discussed above substantially reduces mortality. While similar procedural elements have been described individually in the literature, the present study is, to our knowledge, the first to explicitly report complete elimination of perioperative mortality through systematic refinement of a standard rat permanent LAD ligation protocol. Nevertheless, surgical experience remains a decisive factor, allowing further improvements and, in our optimized protocol, complete elimination of perioperative mortality in the final cohort. When combined with precise anatomical handling, careful anesthesia management, and continuous intraoperative monitoring, these refinements markedly decrease invasiveness and mortality, representing a major advancement that minimizes animal use while ensuring reproducibility and safety in the rat permanent LAD ligation model.

## 5. Conclusions

In conclusion, the present study demonstrates that substantial perioperative mortality is not an unavoidable limitation of the rat permanent LAD ligation model. While similar procedural elements have been described individually, this work is, to our knowledge, the first to demonstrate complete elimination of perioperative mortality through systematic, data-driven refinement of a standard PL protocol. By challenging the prevailing assumption that high mortality is intrinsic to this model, our findings redefine achievable standards for surgical MI induction in rats. When implemented as an integrated workflow, these refinements substantially enhance reproducibility, reduce animal use, and improve animal welfare while preserving the physiological relevance of the model. Together, these improvements ensure higher scientific quality while upholding ethical responsibility in cardiovascular research.

## Figures and Tables

**Figure 1 animals-16-00099-f001:**
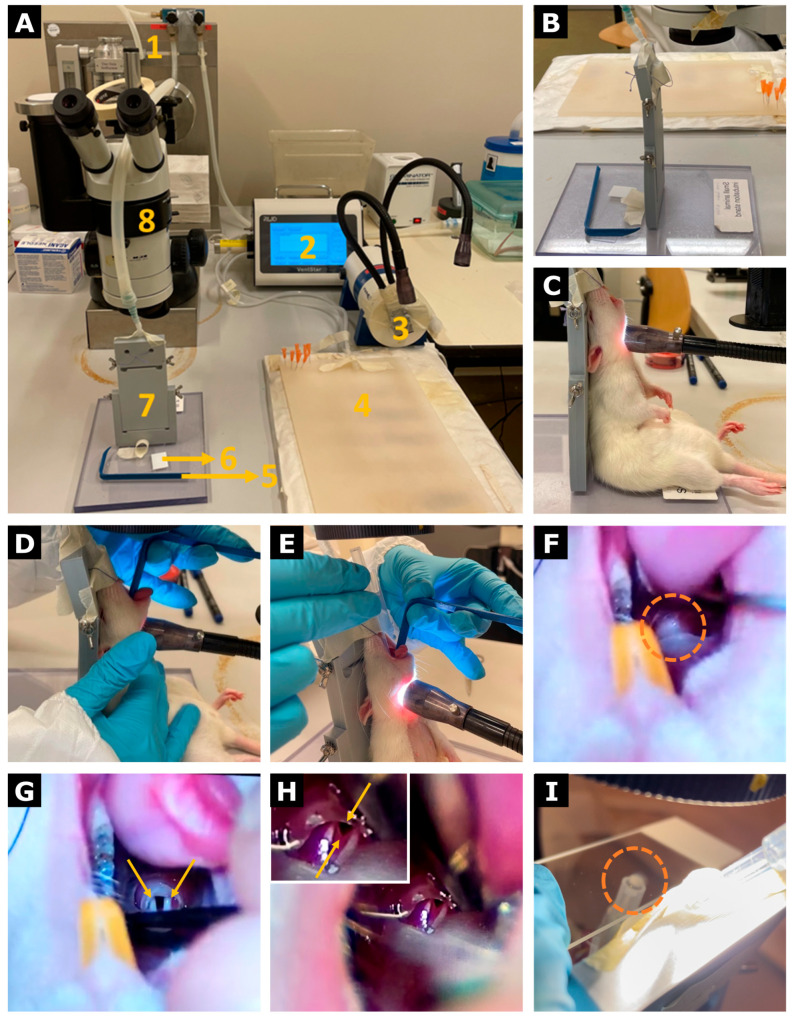
Surgical setup and orotracheal intubation procedure. (**A**) Surgical setup showing the isoflurane vaporizer (1), ventilator (2), snake lights (3), heating pad with a silicone mat (4), laryngoscopic blade (5), microscope slide (6), intubation stand (7), and stereomicroscope (8). Orotracheal intubation is performed under microscopic visualization at 6× magnification. (**B**) Close-up of the intubation stand. (**C**) The rat is positioned on the intubation stand, with the upper incisors secured against the nose cone to maintain inhalation anesthesia. External illumination of the throat allows visualization of the vocal cords and trachea. (**D**) Gentle pressure applied to the neck at the level of the thyroid cartilage improves tracheal visualization. (**E**) A laryngoscopic blade is used to clear the tongue from the field of view. (**F**) Obstructed visualization of the throat (orange dotted circle). (**G**) Gentle advancement of the endotracheal tube opens the oropharynx, exposing the vocal cords (yellow arrows). (**H**) The trachea (yellow arrows) is visible between the vocal cords. (**I**) Correct placement of the endotracheal tube is confirmed by condensation (“fogging”) on a microscope slide during ventilation (orange dotted circle).

**Figure 2 animals-16-00099-f002:**
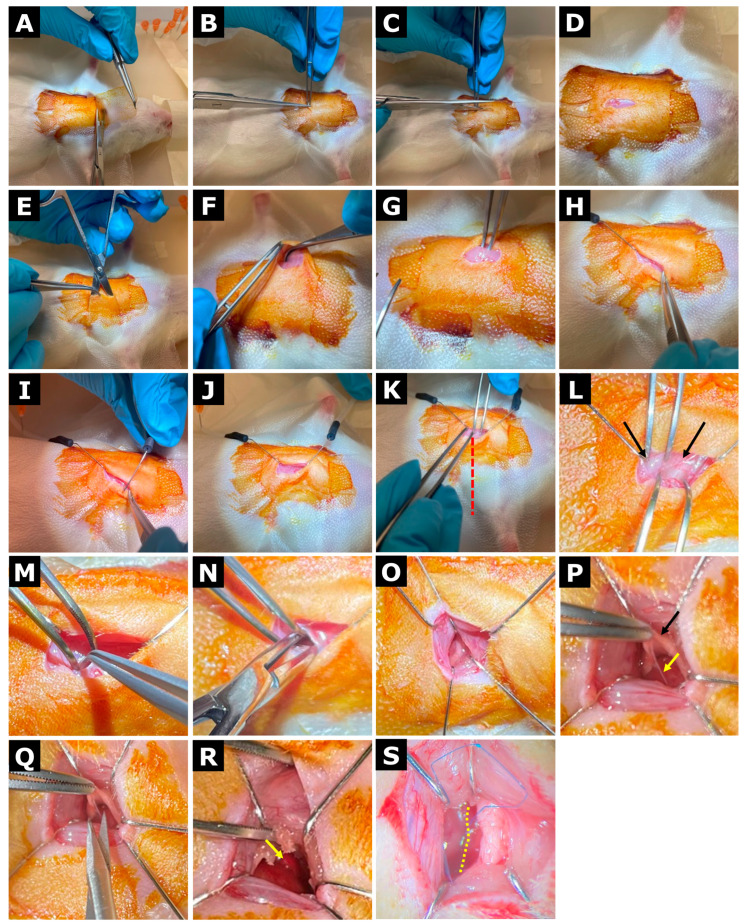
Permanent LAD ligation procedure. (**A**) A surgical window is created in the adhesive drape to expose the operative field. (**B**–**D**) A longitudinal midline incision is made over the sternum. (**E**) A subcutaneous pocket is created by blunt dissection to expose the *Pectoralis major* muscle. (**F**,**G**) The *Pectoralis major* is separated from the intercostal muscles and displaced laterally to the right. (**H**–**J**) The muscle is secured in position using two sharp retracting hooks. (**K**,**L**) The third intercostal space is identified by alignment with the base of the left axilla (red dotted line) and located between the third and fourth ribs (black arrows). (**M**) A left-sided thoracotomy is performed by blunt dissection using anatomical forceps. (**N**) The incision is widened using hemostatic forceps. (**O**) The thoracic cavity is retracted using three to four hooks. (**P**,**Q**) The pericardium (yellow arrow) is opened by gently elevating the thymus (black arrow) to allow incision with micro scissors. (**R**) The left atrial appendage is identified (yellow arrow). (**S**) The LAD is ligated, with successful ligation confirmed by immediate pallor of the left ventricular wall distal to the ligation site (yellow dotted line indicates the border between infarcted and healthy myocardium).

**Figure 3 animals-16-00099-f003:**
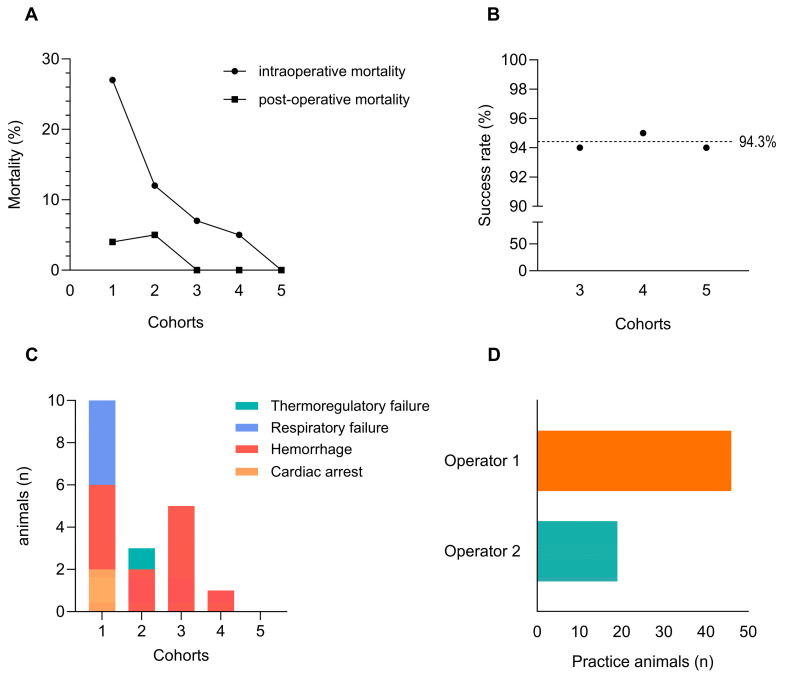
Evaluation of surgical outcomes throughout protocol refinement. Key surgical outcome parameters were recorded during stepwise refinement of the permanent ligation (PL) procedure across five experimental cohorts (cohort 1: *n* = 37; cohort 2: *n* = 25; cohort 3: *n* = 73; cohort 4: *n* = 20; cohort 5: *n* = 17), including: (**A**) intra- and post-operative mortality rates; (**B**) success rate of myocardial infarction induction as verified by echocardiography (subset of cohort 3: *n* = 32; cohort 4: *n* = 18; cohort 5: *n* = 16); (**C**) primary causes of mortality identified during or after surgery. Additionally, (**D**) comparative training effort between two operators is shown, expressed as the number of animals required to achieve consistent surgical success. Operator 1 developed and refined the protocol independently, whereas Operator 2 was trained under supervision using the optimized procedure.

**Figure 4 animals-16-00099-f004:**
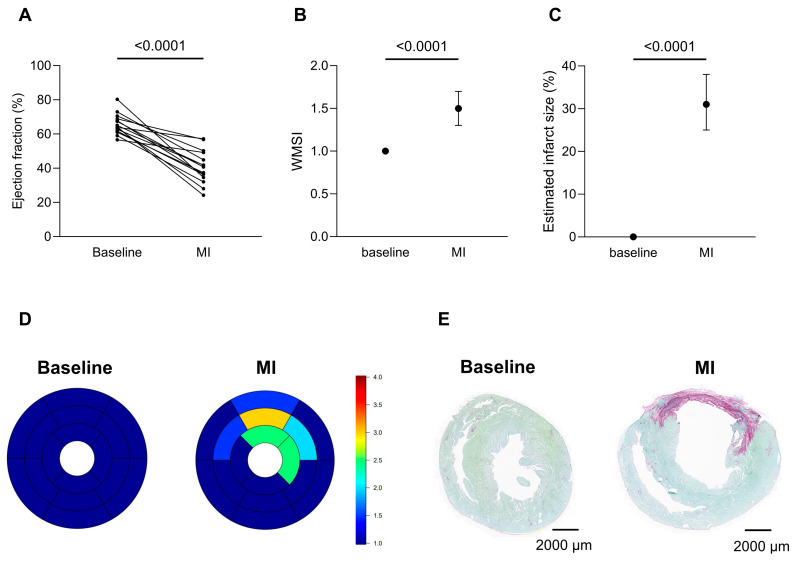
Functional and histological infarct assessment following optimized PL. The extent of MI was evaluated in rats subjected to the fully refined PL protocol (*n* = 15) using functional parameters and histological staining. (**A**) Left ventricular ejection fraction, (**B**) WMSI, and (**C**) estimated infarct size at baseline and four weeks post-PL highlight the functional impact of the infarct. Data are shown as median with interquartile range (IQR); statistical analysis was performed using the Wilcoxon signed-rank test. (**D**) Bull’s-eye plots illustrate regional wall motion abnormalities based on the 16-segment model. Scores range from 1 (blue) to 4 (red) (1 = normokinetic, 2 = hypokinetic, 3 = akinetic, 4 = dyskinetic), with 0.5 increments representing intermediate wall motion abnormalities. (**E**) Sirius Red staining demonstrates the fibrotic scar (red) against healthy myocardium (green). WMSI: wall motion score index, MI: myocardial infarction.

## Data Availability

The raw data supporting the conclusions of this article will be made available by the authors on request.

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
