# Peer review of "A Refined Approach to Permanent Coronary Artery Ligation in Rats: Enhancing Outcomes and Reducing Animal Burden"

_animals, 2025, doi:10.3390/ani16010099_

Round 1

Reviewer 1 Report

Comments and Suggestions for Authors

Myocardial infarction remains one of the most prevalent diseases worldwide. Consequently, the development and optimization of experimental models for myocardial infarction is a crucial task in translational medicine.

The authors have set an important and practical goal: increasing the survival rate of rats following infarction modeling surgery. Achieving this goal would not only adhere to the 3R principles (Reduction in animal use) but also enhance the reproducibility and predictability of research outcomes. It should also be noted that the authors have provided a detailed and clear description of the surgical protocol.

Despite the undoubted importance of the topic, the current version of the manuscript primarily contains general statements and fails to highlight the specific scientific contribution of this work. To demonstrate the fundamental difference of this study from previous works, such as the study by Ellen Heeren et al., the following additions and revisions are recommended:

  1. Techniques such as tracheal intubation, inhalational isoflurane anesthesia, and the permanent ligation (PL) model are currently widely used. What constitutes the unique nature and key improvementof the protocol proposed by the authors? It is necessary to clearly articulate what specific modifications were introduced and how exactly they contribute to increased survival.
  2. Controlling body temperature and having the surgery performed by an experienced surgeon are undoubtedly established standards in surgical research. While their importance cannot be overstated, these points in themselves do not represent scientific novelty. The emphasis should be shifted to how the adherence to these standards within the proposed protocolled to unique outcomes.
  3. To strengthen the scientific substance, it is essential to add an analysis of the phenomenon of anesthetic preconditioningto the Discussion section. The cardioprotective effect induced by the combination of isoflurane and oxygen may directly influence the increase in animal survival. Discussing this factor and how the authors accounted for it in their study would significantly deepen the interpretation of the obtained data.
  4. Incorporating electrocardiography (ECG)data and other functional tests would significantly strengthen the evidence base. The availability of such data would allow not only for the confirmation of survival but also for an objective assessment of the success of infarction modeling and the post-operative functional state of the heart.

Conclusion
In its current form, the manuscript reads more like a description of a standard protocol. To present the work as a significant scientific study, the text needs to be revised to emphasize the innovative aspects of the methodology and supplemented with an in-depth discussion of the results in the context of current literature.

Author Response

Reply to reviewers

We would like to thank the editor and reviewers for the critical evaluation of our manuscript and for giving us the opportunity to revise it. We replied to the reviewer’s comments point by point and based on the constructive remarks of the reviewer, adjusted the manuscript accordingly. Modifications in the manuscript were highlighted in yellow.

Point-by-point response to the Reviewer’s 1 comments

As pointed out by Reviewer 1, we fully acknowledge that, in its initial form, the manuscript did not sufficiently highlight the specific scientific contribution and methodological innovation of our work. In response to these comments, we have substantially revised the manuscript, with particular emphasis on the Introduction and Discussion sections, to better clarify the fundamental differences between our study and previously published protocols and to more clearly articulate the novelty of our approach.

Below, we provide a point-by-point response to the reviewer’s comments.

1-Techniques such as tracheal intubation, inhalational isoflurane anesthesia, and the permanent ligation model are widely used. The unique nature and key improvements of the protocol must be clearly articulated.

We fully agree with this comment. In the revised manuscript, we now explicitly clarify that the novelty of our work does not lie in the introduction of individual techniques, but rather in their systematic and integrated implementation within a standardized refinement strategy aimed at eliminating perioperative mortality.

Specifically, in the revised manuscript, we emphasize that the protocol combines several interdependent refinements, including standardized orotracheal intubation under exclusive inhalation anesthesia, strict intraoperative temperature control, an optimized medial thoracotomy approach, careful tissue handling, continuous monitoring of cardiac rhythm and appropriate corrective action, reference-ligature–guided identification of the LAD, and carefully timed extubation.

Importantly, these refinements were not introduced simultaneously, but were prospectively evaluated across five successive cohorts (n = 172). This iterative approach allowed us to identify specific causes of perioperative mortality and to eliminate them stepwise. As a result, perioperative mortality was reduced to 0% in the final optimized cohort, a result that, to our knowledge, has not previously been reported for a standard rat permanent LAD ligation model.

We now clearly state that the key advance of this study is the integrated optimization of the entire surgical workflow, rather than incremental technical modifications, and that perioperative mortality should be considered a modifiable experimental outcome, rather than an inherent limitation of the model as often mentioned in the literature.

2- Temperature control and experienced surgeons are established standards and do not represent novelty; the focus should be on how adherence to these standards led to unique outcomes.

We fully agree and have revised the manuscript accordingly. In the Discussion section, the emphasis has been shifted from the standards themselves to their quantified impact on experimental outcomes.

In particular, we now show that improved airway management and standardized thoracotomy approach eliminated respiratory-failure mortality observed in cohort 1. Likewise, careful positioning and continuous adjustment of retracting hooks prevented mechanical restriction of cardiac movement, thereby avoiding procedure-induced bradycardia and cardiac arrest. Additionally, continuous temperature monitoring (maintained between 35–38 °C), in combination with the use of an optimized surgical drape, was associated with elimination of hypothermia-related mortality observed in cohort 2. We demonstrate that structured, supervised training using the optimized protocol substantially reduced the learning curve. Operators trained under supervision required approximately 59% fewer animals to achieve consistent surgical success, directly contributing to refinement and reduction within the 3Rs framework.

These elements are now explicitly linked to the observed improvements in survival and reproducibility, rather than being presented as general best-practice recommendations.

3- An analysis of anesthetic preconditioning should be added, as the combination of isoflurane and oxygen may contribute to cardioprotection and increased survival.

We thank the reviewer for this important suggestion. A dedicated paragraph on anesthetic preconditioning has now been added to the revised Discussion section. We acknowledge that volatile anesthetics such as isoflurane are known to induce cardioprotective preconditioning through mitochondrial and pro-survival signaling pathways. We further discuss how the standardized use of isoflurane, in combination with controlled ventilation and oxygen delivery, may have contributed to improved perioperative stability and survival in our model. At the same time, we clearly state that the present study was not designed to mechanistically investigate anesthetic preconditioning. Its potential contribution is therefore discussed transparently and cautiously. Importantly, we emphasize that the observed survival benefit cannot be attributed to anesthetic choice alone, but rather to its integration with reliable airway management, continuous monitoring, controlled ventilation, and carefully timed extubation.

4- ECG data and functional tests would strengthen the evidence by confirming infarction success and post-operative cardiac function.

While intraoperative ECG recordings were not collected, we have clarified in the Results section that infarction success and functional outcome were robustly validated using complementary approaches. Longitudinal echocardiography performed in the optimized cohorts demonstrated a significant reduction in left ventricular ejection fraction (median ~40%) 4 weeks post-MI. In addition, regional wall motion abnormalities were quantified using the 16-segment wall motion score index (WMSI), providing objective confirmation of LAD-territory infarction. Finally, infarct severity was assessed both functionally and histologically, with Sirius Red staining demonstrating consistent scar formation. Together, these data provide strong functional and structural confirmation of successful infarction modeling beyond survival alone.

In summary, we believe that the manuscript has been substantially revised to fully address the reviewer’s concerns and to reposition the work from a descriptive protocol toward a methodologically rigorous, data-driven refinement study, important in the current context of 3R’s. We sincerely thank the reviewer for his/her constructive and insightful comments, which have significantly improved the clarity, rigor, and scientific impact of our manuscript.

Reviewer 2 Report

Comments and Suggestions for Authors

Dear authors,

thank you very much for submitting your manuscript. Please find the Reviewer's Report down below.

Kind regards.

Reviewer's Report:

Summary:

In this study the authors examine different refinements in a rat model for coronary artery ligation. Especially procedures of intubation, regulation and monitoring of body temperature as well as the training of the operator were used and analyzed. They were able to show significant on mortality rates and animal welfare.

General comment on the hypothesis of the work:

The present study is very interesting and provides important information and procedures to improve the rat model for coronary artery ligation.

When listing the authors' institutions, the same institution is mentioned five times. All authors belong to one institution, so it only needs to be mentioned once. Email addresses can be listed below.

The simple summary is short and precisely written.

In the abstract the methods and the results of the study are described in detail. But the number of rats included in the experiments are missing.

The introduction is too superficial and needs more information on previous methods used in rat models for coronary artery ligation. In addition, the animal species are often missing from the references cited. In its current form, the introduction consists of three separate paragraphs that are barely related to each other. This needs to be revised and formulated more fluently.

In the Materials and Methods section, further additions must be made to materials and methods used. See detailed comments down below.

In the results the section 3.1 is a repetition of the materials and methods described. Only Figures 1 and 2 have been added as expanded illustrations. This section should be integrated into the Materials and Methods section. The results of the method used are described starting in Section 3.2.

The discussion is very detailed and well written.

The references must be revised according to the guidelines of the journal.

Comments on Introduction:

L56: Please write out the abbreviation bpm.

LL62-65: Please explain the species. Did you mean human?

L74: Mortality rates in which species?

Comments on Material and Methods:

Please check the formatting according to the template of the journal.

L86: Please add the manufacturer, city, federal and national state for the hot bead sterilizer.

L90: Please add the manufacturer, city, federal and national state for the heating pad.

L91: Please add the manufacturer, city, federal and national state for the silicone mat.

L92: Please add the manufacturer, city, federal and national state for the 25G needles.

L94: Please add the manufacturer, city, federal and national state for the stereomicroscope, isoflurane delivery system and snake lights.

L97: Why did you use only female rats for the experiment? Please add the total number of participants.

L100: Please add the manufacturer, city, federal and national state for the induction chamber and the isoflurane.

L112: Please add the manufacturer, city, federal and national state for the rectal probe.

L118: Please add the manufacturer, city, federal and national state for the disinfectants.

L119: Laval, Canada

L126, 128: Pectoralis major (capital letter, italic style)

L40: Please add the manufacturer, city, federal and national state for the 8-0 Prolene suture.

L147: Please add the manufacturer, city, federal and national state for the 5-0 Vicryl suture.

LL148-149: Pectoralis major (capital letter, italic style)

L150: Please add the manufacturer, city, federal and national state for the disinfectant.

L166: Please add the manufacturer, city, federal and national state for the type IV cage.

LL201-203: Please add the statistical program used with manufacturer, city, federal state and state. Please describe the significance levels.

Comments on the Results:

LL239, 240, 267, 268: Pectoralis major (capital letter, italic style)

 Comments on the discussion:

L352: Pectoralis major (capital letter, italic style)

LL379-380: Are there prevalences of tracheal injuries of other or common procedures of extubation in rats? Please discuss the current literature.

LL402-405: Please add a reference.

L417: Kainuma et al. [28]

Author Response

Reply to reviewers

We would like to thank the editor and reviewers for the critical evaluation of our manuscript and for giving us the opportunity to revise it. We replied to the reviewer’s comments point by point and based on the constructive remarks of the reviewer, adjusted the manuscript accordingly. Modifications in the manuscript were highlighted in yellow.

Point-by-point response to the Reviewer’s 2 comments

We thank the reviewer for the thorough evaluation of our manuscript and for the constructive comments. We appreciate the positive assessment of the relevance of our study and its contribution to improving the rat coronary artery ligation model with respect to mortality reduction and animal welfare. All comments have been carefully considered and addressed in the revised manuscript, as outlined below.

1- General comments

  • The institutional affiliations have been corrected, as all authors belong to the same institution.
  • The total number of animals included in the study has been added to the Abstract.
  • The Introduction has been revised to provide a more coherent and fluent overview of previously published rat coronary artery ligation models
  • Appropriate manufacturer information has been added in the Material and Methods section.
  • Section 3.1 of the Results, which largely reiterated procedural aspects, has been integrated into the Materials and Methods section. The Results section now focuses on outcome-related data.
  • The reference list has been revised and formatted according to the journal guidelines.

2- Comments on the introduction, material and methods, results and discussion

The Introduction was restructured to provide a more coherent overview of existing rat coronary artery ligation models, clearly specify the animal species, and better position the novelty of the present work. The Materials and Methods section was expanded and corrected to include full methodological details, manufacturer information, rationale for animal selection, and statistical analyses, while procedural descriptions previously repeated in the Results were integrated here. The Results section was adjusted to focus exclusively on outcome data, including survival, functional validation, and infarct characterization. The Discussion was refined to better contextualize the findings within the current literature, including anesthetic preconditioning, respiratory failure-related complications, and the impact of procedural refinements on mortality, reproducibility, and animal welfare.

We believe that these revisions have improved the clarity, structure, and completeness of the manuscript. We thank the reviewer again for the valuable comments, which have helped to strengthen the revised version of our work.

Round 2

Reviewer 1 Report

Comments and Suggestions for Authors

Colleagues, the amendments provided clarification on the methodology and application of the model in scientific practice.
I believe the amendments fully reflect the intention of the work and provide an opportunity to draw on research from related fields to utilize the data in the scientific article.
The article may be accepted in its current form.

Reviewer 2 Report

Comments and Suggestions for Authors

Dear authors,

thank you very much for incorporating my comments. The abstract is now easier for the reader to understand. The introduction now reads much more smoothly. Important aspects have been added to the Materials and Methods section, and the now apparent separation of methods and results gives the manuscript more structure. However, the numbering of the Materials and Methods section needs to be adjusted again (2. Materials and Methods; 2.1; 2.2, etc.).
The results and discussion are well presented.

Kind regards.